# Combined Oxygen–Ozone and Porcine Injectable Collagen Therapies Boosting Efficacy in Low Back Pain and Disability

**DOI:** 10.3390/diagnostics14212411

**Published:** 2024-10-29

**Authors:** Manuela De Pascalis, Susanna Mulas, Liliana Sgarbi

**Affiliations:** Division of Rehabilitation, ASST Fatebenefratelli-Sacco Hospital, Fatebenefratelli Institution, 3, Piazza Principessa Clotilde, 20121 Milan, Italy; manuela.depascalis@asst-fbf-sacco.it (M.D.P.); susanna.mulas@asst-fbf-sacco.it (S.M.)

**Keywords:** low back pain, oxygen–ozone therapy, porcine collagen type 1, porcine collagen injection, low back pain clinical management, combination therapy, numerical classification scale, Roland–Morris questionnaire

## Abstract

**Background/Objectives:** Intervertebral disc degeneration is the most common cause of low back pain (LBP), and lumbosciatica is a major challenge to healthcare systems worldwide. For years, ozone therapy has been used with excellent results in intervertebral disc disease and in patients with LBP. In vitro studies have demonstrated the positive action of porcine collagen in extracellular matrix remodeling and homeostasis. These tissue changes, associated with LBP, may suggest an indication for combined ozone/collagen treatment in patients with LBP. However, no studies have been reported regarding this combination of treatments. **Methods:** The present work compared retrospective data of two treatment groups (each of 10 LBP patients): (A) oxygen–ozone therapy (OOT) vs. (B) OOT plus porcine collagen type 1 injections (COL I). Pain intensity and physiological function were assessed by the numerical rating scale (NSR) method. The Roland–Morris questionnaire was used to assess disability. Patient data were acquired before, during, and at the six-month follow-up. Significant differences were assessed by ANOVA and Student’s *t*-test. **Results:** The analyses revealed significant statistical differences comparing the two arms, where the (OOT+COL I) treatment demonstrated a booster efficacy in pain (a reduction of 62% vs. 35%), while the questionnaire revealed a reduction in disability (70% vs. 31%). **Conclusions:** Therefore, this combination therapy (oxygen–ozone plus porcine injectable collagen) might be a promising approach for the management of patients with LBP.

## 1. Introduction

Low back pain (LBP) is a common disorder with a significant impact on patients in real life and their clinical behaviors; however, it has a major impact within the socioeconomic community and healthcare facilities [1]. The prevalence of this disease is estimated to be between 22% and 65% per year and can reach up to 80% of the population, where LBP disease can have mild to severe manifestations [1]. In about 60–80% of cases, no specific cause is diagnosed and the perceived pain is attributed to muscle or ligament tensions and only in 5–15% of cases are the causes of pain associated with degenerative phenomena and disc injuries [1]. A herniated disc with symptoms is a degenerative disease of the intervertebral disc, which can present with low back pain, lumbosciatica, or lumbocruralgia due to root compression [2]. However, lumbar disc herniation (LDH) is also frequently detected in asymptomatic individuals who undergo further diagnostic tests for other medical disorders and its prevalence is estimated at 57% [3]. LDH is, therefore, a common condition that, when symptomatic, has a prognosis that is not always favorable with a tendency towards chronicity [4]. Therefore, considerable effort has been made to identify the most effective way to combat this condition in order to support and improve patients’ conditions [5,6]. Recent guidelines suggest a different technical approach in order to combat, treat, and cure LBP disease [7]. The American Society of Pain and Neuroscience (ASPN) [7] has indicated several procedures and/or treatments, including in the fight against LBP. The armamentarium of treatments includes anti-inflammatory therapy [8], ozone treatment [9,10,11], minimally invasive procedures [12,13], regenerative drugs [7,14,15], and surgery [16,17]. However, one treatment that has proven to be very effective in treating this condition has been ozone treatment [18,19]. The effectiveness of oxygen–ozone therapy in medicine is now well-defined and demonstrated in various fields, such as vascular diseases, orthopedics, and dentistry [10]. The rationale for the use of oxygen–ozone infiltrative therapy (OOT) in the treatment of low back pain due to disc disease is based on the combination of the anti-inflammatory action with the action of accelerating the process of the dehydration of the cartilage tissue of the disc. In particular, ozone oxidizes water-rich mucopolysaccharides, resulting in the dehydration of the protruding or herniated material [18]. The anti-inflammatory action is instead related to the oxidative capacity of the carbon–carbon double bond of arachidonic acid with a consequent reduction in the production of prostaglandins [19]. At the same time, the reactivation of the microcirculation facilitates the elimination of pro-inflammatory mediators [20].

In non-pathological discs, nerve afferents are limited to the outer third of the disc and are not found in the inner ring or the nucleus pulposus region [21,22]. In contrast, in pathological discs, nociceptive nerve fibers along with vascular segments may migrate into the central regions of the disc [23,24]. It is hypothesized that neurotransmitters along with changes within the extracellular matrix (ECM) itself and the presence of cytokines act on the nervous part of the disc. In addition, pain-related peptides and pro-inflammatory cytokines increase at this stage [21,23].

This explains the role of disc structures in low back pain and why, in patients suffering from chronic low back pain, the fibers of the connective tissue have a different orientation than in healthy subjects. This is evident in the study of Langevin et al. [25] (an ultrasound comparison between perimuscular connective tissue of the lumbar region in a group of subjects without low back pain and a group of subjects with chronic or recurrent low back pain for more than 12 months), demonstrating that the group with chronic low back pain had a perimuscular thickness 25% greater than normal subjects. The connective tissue of subjects with chronic low back pain appears disorganized, remodeled with infiltrations of adipose tissue and signs of fibrosis [26]. It is not yet possible to establish whether the observed tissue changes are the cause or consequence of chronic low back pain, but it is possible to provide the necessary substrate for favorable modernization. Tenocytes are specialized fibroblasts within the connective tissue, responsible for the remodeling of the extracellular matrix (ECM) influencing type I collagen turnover mechanisms (COL-I), the main component of the ECM [27,28]. The tendons are placed between the muscles and the bones and transfer the forces generated by muscle contraction to the skeleton [29]. In fact, porcine collagen type 1 is able to stimulate the contraction of fibroblasts that generate and exert forces on the ECM through the contraction itself. An optimal level of the contractile capacity of fibroblasts is necessary in order to increase the tensile strength and facilitate the repair phenomena. Tropocollagen (consisting of three alpha helices) is the basic functional unit of mature collagen and represents the substrate necessary for the regeneration of collagen fibers. In fact, two different recently published studies have demonstrated the effectiveness of porcine collagen type 1 (COL I) supplementation in both the synthesis and migration [30] and regeneration of the collagen structure in connective tissues [31]. Nonetheless, the efficacy and tolerability of porcine collagene type I was assessed previously [32,33]. To our knowledge, there are no studies on LBP patients treated simultaneously with OOT and COL I to date. Indeed, in this study, the OOT treatment was compared with the combined action of OOT+COL I in patients affected by LBP and lumbosciatica, evaluating the improvement of pain, the functional statement, and the level of disability.

## 2. Materials and Methods

### 2.1. Patients

The study was conducted from September 2022 to November 2023. Data collection was performed retrospectively, preserving the anonymity of patients’ personal data. Fifty-seven consecutive patients affected by LBP and lumbosciatica were screened in our institution. Using the formula for sample size evaluation (power study: 85% and adverse event: 10%), we obtained the N value (number of subject) equal to 18. Indeed, a total 20 patients out of 57 (35.08%; 10 males and 10 females) were enrolled in order to receive treatments, (mean 58 years; CI 38:79). The remaining 37 patients had at least one of the exclusion criteria.

The exclusion criteria included the following: acute radicular signs in the lower extremities, oncological pathologies in the active phase and/or under investigation, cognitive impairment, and patients reluctant to give informed consent.

The inclusion criteria included the following: lumbosacral MRI with evidence of herniated disc or multiple disc protrusions, persistent low back pain for at least six months, NSAID and painkiller therapy discontinued for at least two weeks, and cortisone therapy discontinued for at least two months. Patients suffering from low back pain (value of numerical rating scale >4.0) for more than six months who underwent MRI of the lumbosacral spine and evidence of disc disease were included in the study. Patients were divided into two equal groups: group A and group B. Each treatment arm (A and B) comprised 5 males and 5 females (Table 1). While, patients with LBP and lumbosciatica were 5 vs. 5 and 4 vs. 6 in the group A and B, respectively).

### 2.2. Treatments

All patients (A and B) had been treated with intramuscular paravertebral injections of O_2_O_3_ (concentration equal to 10 μg/mL; OOT treatment). The total volume administered corresponds to 20 mL divided into four injection sites (5 mL per injection site), through the 32 G needle (size: 0.7 mm × 32 mm). All patients underwent treatment twice a week for a total of 8 consecutive infiltrative treatments (one month). Patients included in group B additionally received intramuscular administration of a vial containing 2 mL of porcine collagen type 1 (MD-LUMBAR, Guna, Milan, Italy; COL I) divided into 4 injection sites (0.5 mL per injection site), through a 27 G needle (size: 0.4 × 19 mm).

Collagen injection was performed after 20 min of lumbar administration of O_2_O_3_ (Figure 1). In order to select the injection point, the physicians evaluated the interest point by MRI image evaluation. The injection point of patient was evaluated through palpation to locate the boundary space between two vertebrae (i.e., L5 and S1).

### 2.3. Data Collection

All data were acquired at the following time points: T0 (patient enrollment), T1 (after one month of treatment), T2 (end of treatment), and T3 (six months after treatment). The level of pain in LBP patients were obtained through a numerical rating scale (NRS; range of 0–10; 0 = no pain and 10 = full pain). In order to assess treatment-associated outcomes, the Roland-Morris Disability Questionnaire (RMD-Q) was used. The level of disability was established through the responses acquired by 24 different questions. A binomial evaluation (0/1; 0 = ability; 1 = inability) was possible for each. The total score of the questionnaire was obtained by summing the points attributed by the patients. The total result can range from 0 (equals no disability) to 24 (equals heavy inability). In fact, the reduction in the total score of RMD-Q means an improvement in physiological function. The reduction in NRS score was interpreted as a positive outcome after treatment. The delta variation of NRS were obtained by comparing the follow-up points (T1, T2, and T3) with respect to the T0 point (starting point).

Functional improvement was also assessed. Forward flexion and lateral flexion (right and left) were quantified in cm, measuring the distance from the fingertips to the ground, at the moment of maximum flexion uttered by the patient. The Shapiro–Wilk tests were performed in order to understand whether we have a normal distribution of data. Here, the result concerning both W and *p* values for the OOT group (W = 0.9066 and *p* = 0.2646) and OOT+COL I group (W = 0.9175 and *p* = 0.3476) are reported, respectively.

The analyses were performed via the GraphPAD software 8.0 (San Diego, CA, USA), using both ANOVA and Student’s test. Statistically significant values were considered when *p* values were <0.05.

## 3. Results

### 3.1. NRS Analysis

All patients were assessed with the NRS scale at time T0 before starting treatment. The mean pre-treatment NRS score was 6.5 ± 1.95 and 6.8 ± 1.53 for group A and group B, respectively. No statistical differences were observed between the two groups at the beginning of the protocols (Figure 2A). Similarly, the positive reduction in NRS was observed in two groups (Figure 2B,C), where the mean values were 3.9 ± 2.55 and 3.8 ± 2.50 for group A and group B, respectively. At these two checkpoints, we can assume that there were no statistical differences by comparing the two treatments. However, comparing the follow-up breakpoint, we observed statistical differences in terms of the mean value of NRS (Figure 2B, *p* < 0.001) and their delta change of the cognate NRS over time (Figure 2D; *p* < 0.001). In particular, after the treatment, we obtained a reduction of 5.2 points in the NRS scale (average value) in the OOT+COL I treatment instead of 3.6 points (NRS scale) in the OOT treatment alone. However, the percentage difference in reduction was 55.385% and 76.471% in group A and B, respectively. A total of 21% was present between the two treatment arms (Figure 2D). As a final concept, we carefully observed the delta NRS difference between the two treatment groups at point T2, where the changes in the NRS value (expressed as the improvement of the delta point per month) are zero in OOT and 1.4 in OOT+COL I. In addition, at the six-month follow-up, the effect of the addition of COL I treatment seems to better preserve the results obtained with OOT alone (Figure 2B,D).

### 3.2. Functional Improvements: Forward Bending, Lateral Flexions (LF), and Level Disability (Roland and Morris Questionnaire)

Function improvement analyses show a positive trend in the OOT+COL I treatment compared to those observed in the OOT treatment alone (group B). The difference observed in the forward flexion analyses reveals that the OOT+COL I treatment (group B) had the mean distance values between all three visits. In particular, we observed an antiparallel trend by comparing the two groups. The OOT arm increased its main values during the visit, while the OOT+COL I treatments decreased them (*p* < 0.0001). The result could indicate that the OOT+COL I treatment seems to be more effective in the short term and in the long term. In fact, the distribution of data could indicate that the OOT+COL I treatment also maintained its effect as a booster and better maintenance, compared to the OOT treatment (Figure 3A). Similarly, the OOT+COL I treatment (group B) clarified a level of superiority over the OOT treatment (Figure 3C) with regard to the disability questionnaire. Thus far, we would like to mention that the decrease in RMD-Q represents a successful treatment. In the red columns, we can see that the level of disability decreases between visits, while the same parameter in the green columns (OOT treatment only) remains stable after the second visit (*p* < 0.001). In fact, comparing the average value per visit, we observed at least 5.0 points of difference between the average values (Figure 4C). Looking for the lateral, right (Figure 3B), and left (Figure 3D) flexion, no significant differences were obtained from the ANOVA analyses between the OOT and OOT+COL I treatments (*p* > 0.05). However, the results regarding patients included inside group B (Figure 3B,D) showed better results. Overall, we can assess that no inferiority level of the OOT+COL I treatment could be accepted. Movement analyses in the OOT+COL I groups revealed that the patients’ body trunk appears to be more flexible, particularly when looking for the forward flexion parameter.

### 3.3. Functional Improvements: Analyses by Pathologies

The analyses of functional improvement were carried out by splitting the patients’ treatments according the two pathologies investigated (LBP and lumbosciatica; Figure 4) and their associated treatments. The following parameters—forward bending (Figure 4A), lateral flexion to the right (LFR; Figure 4B), disability questionnaire (Figure 4C,D), and lateral flexion to the left (LFL) were evaluated, reporting the delta differences between visit T2 and visit T0 (T2–T0). Negative values were considered a positive factor for the effectiveness of the treatment. For each graph, the global analyses avoiding the type of pathology were reported. These analyses were performed following the same composition of Figure 4. In addition to Figure 4, significant statistical differences were observed in Plots A and B (*p* < 0.001), whereas no statistically significant difference was observed in both plots B and D (*p* = NS). However, all analyses revealed a higher positive trend (more negative values) of OOT+COL I treatment in all parameters analyzed (Figure 4A–D). Specifically, we observed the following:The OOT+COL I treatments increased forward flexion in patients with LBP compared to those with lumbosciatica. The negative values associated with the OOT COL I treatment were higher in patients with LBP (positive effects equal to 70.71%) than those observed in patients with lumbosciatica (positive effects equal to 30.04%; *p* < 0.001), while the OOT treatment alone did not statistically significantly modify forward flexion in either LBP or lumbosciatica patients.The OOT+COL I treatments better improved the disability condition level in patients with lumbosciatica compared to those with LBP. Here, the negative values associated with the OOT COL I treatment were higher in patients with lumbosciatica (positive effects equal to 68.93%) than those observed in patients with LBP (positive effects equal to 56.25%; *p* < 0.001). Notably, OOT treatment alone also affected positively the disability condition; however, the OOT+COL I treatment showed better results.However, the OOT treatment alone did not statistically significantly modify forward flexion in either LBP or lumbosciatica patients. OOT+COL I treatment improved disability in both pathologies (*p* < 0.01)No statistically significant differences were observed in the lateral flexion analyses. However, the LFR results analyzed returned a probability value very close to significance (*p* = 0.0606).Thus far, even in these analyses, non-inferiority levels have been achieved for both pathologies investigated. Indeed, in our personal impression, the combined approach (OOT+COL I) acts as a booster, improving the clinical condition of both LBP and lumbosciatica patients.

## 4. Discussion

This study analyzed 20 patients with LBP, comparing the effect of the OOT+COL I treatment to OOT alone. Usually, the physicians assessed the resolution of pain status, using (mostly) VAS ratings instead of the Likert algorithm Numeric Pain Rating Scale (NRS) [7]. In the present study, we evaluated pain as the first parameter according to the NRS scale. Our result showed much better results comparing OOT+COL I (group B) to OOT alone (group A), looking for pain resolution, improved movement, and reduced disability condition of patients. The arm containing patients treated with OOT+COL I revealed better results than those obtained in the OOT arm alone, within all the parameters investigated. In particular, significant statistical differences were found by analyzing the following: the evolution of pain (*p* < 0.0001), the forward flexion parameter (*p* = 0.001), Figure 2, Figure 3 and Figure 4), and the reduction in disability (Figure 3C and Figure 4C). In addition, the OOT+COL I treatment also positively influenced LPB patients and lumbosciatica patients.

The combination treatment (OOT+COL I) seems to play a fundamental role in improving pain and the physiological condition in the short and long term (six-month follow-up). In any case, the levels of the absence of inferiority were evaluated, taking into account all the parameters investigated. The combination of these treatment regimens (OOT+COL I) has not been evaluated previously in LBP patients and, therefore, we have no matching parameters. The combined action of the injective medical device (MD-LUMBAR) and intra-muscular injection therapy with O_2_O_3_ seems to confirm a booster effect in patients suffering from LBP and lumbosciatica. It is reasonably valid to assume that the combined treatment can act as a booster by simultaneously promoting the anti-inflammatory effect of ozone treatment [34] and the regenerative effect of collagen [30]. This approach could also be called “regenerative medicine”, according to the criteria of the American Society of Pain and Neuroscience (ASPN) guidelines [7]. It is important to understand that not all biologics used in regenerative medicine are equivalent. The patient’s health status and comorbidities, the medications the patient takes, and the parameters and the protocol used for cell collection influence or may influence the final result of mesenchymal stem cell (MSC) collection. Because these variables could not be fully controlled, clinical trials evaluating regenerative medicine for LBP in discogenic disease, including prolotherapy, protein-rich plasma (PRP), cell therapy, and other intra-disc injections, were reviewed [7]. The ASPN guidelines do not mention the use of porcine collagenase in the regenerative medicine chapter; however, the action of porcine collagen type 1 affects collagen turnover [30,31], and its safety and efficacy [32,33] were reported in previous studies [32,33]. The efficacy of COL I medical device might probably be associated with the connective tissue structure, characterized by the presence of the extracellular matrix (ECM). [30] The ECM, present in all tissues, represents the substrate on which tissue cells can adhere, migrate, proliferate, and differentiate. It consists of macromolecules (proteoglycans and specialized proteins such as elastin and fibronectin) that influence tissue cell functions, indirectly controlling the physiological, pathophysiological, and pathological phenomena. The connective tissue is structurally characterized by the presence of the extracellular matrix (ECM) [31]. The ECM, present in all tissues, represents the substrate on which tissue cells can adhere, migrate, proliferate, and differentiate. It consists of macromolecules (proteoglycans and specialized proteins such as elastin and fibronectin) that influence tissue cell functions, indirectly controlling the physiological, pathophysiological and pathological phenomena [35]. Considering these reasons, the results of the present study confirm the positive effects of the OOT COLI treatment demonstrating the efficacy on both short-term and long-term painful symptoms. The COL I medical device strongly boosts the action of ozone, establishing a metabolic synergism against LBP and lumbosciatic diseases. In addition, COL I improves the motor function of the spine by demonstrating improved forward flexion and improves the overall disability condition of patients with both low back pain and lumbosciatica.

Although the treatments with added porcine collagen type 1 improve the lateral flexion of the spine (right and left) overall, the data do not indicate a significant difference compared to the treatments with OOT alone. The addition of porcine collagen type 1, by injection, can open three interesting points of reflection:(1)The self-diffusion capacity of the medical device (MD-LUMBAR) highlights the ability of collagen molecules to enrich the site of interest, where the EUS-guided injection could be a non-mandatory approach but still remain recommended. In particular, some authors have demonstrated through their results, in the treatment of shoulder pathologies, that the non-use of the ultrasound-guided approach in order to perform an infiltrative treatment showed better results than the ultrasound-guided approach [36];(2)The combined action of OOT+COL I increased the effect of the therapy, as a booster, in patients affected by LBP and lumbosciatica;(3)At the same time, the regenerative properties of porcine collagen type 1 could lead to this treatment being included within the branch of regenerative medicine.

Although the results of the present study strongly suggest a benefit for patients with LBP and lumbosciatica as well, the results did not report any confirmation of the modification of the patient’s morpho-physiological conditions. Furthermore, it is absolutely necessary to emphasize that the group of patients with lumbosciatica is only nine in total, and, although statistical tests revealed statistically significant differences, these results need to be confirmed in a larger cohort of LBP patients with lumbosciatica. However, the indications of the Italian health system avoid an MRI investigation, in the absence of symptoms. Furthermore, this aspect is also underlined in the guidelines for non-surgical treatment, in which the authors confirmed the non-need for MRI imaging, if the symptoms are not present (PICO 4 point) [37]. Certainly, the total number of patients analyzed is not very large; however, the sample size analyzed was found to be sufficient. Further studies with higher sample sizes will be necessary for the possible confirmation of the results presented in this study.

## 5. Conclusions

Previously, no studies had been conducted regarding the combined use of OOT and porcine collagen type 1 injection. The combination of these two injective therapies has been shown to be safe, feasible, and effective. No adverse events were found in any patient. Therefore, the addition of porcine collagen I treatment to OOT has been shown to boost the results obtained with the ozone-based treatment alone in the short and long term. Further investigation including a large number of patients might be useful in order to confirm the results in this proof-of-concept study.

## Figures and Tables

**Figure 1 diagnostics-14-02411-f001:**
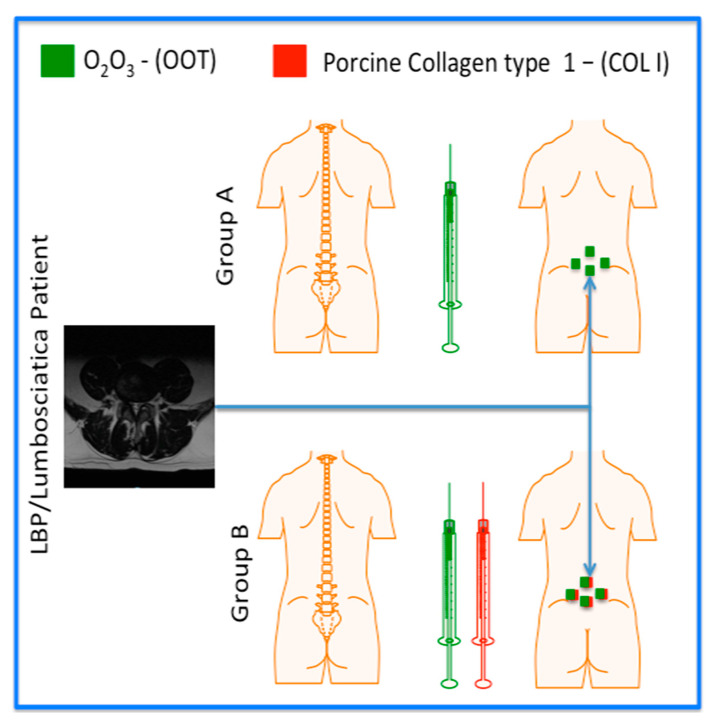
Schematic representation of the subdivision of the two groups (A and B) treated with oxygen–ozone treatment (OOT) and porcine collagen type 1 (COLI). To the left, MRI drives the physician to select the points of injections.

**Figure 2 diagnostics-14-02411-f002:**
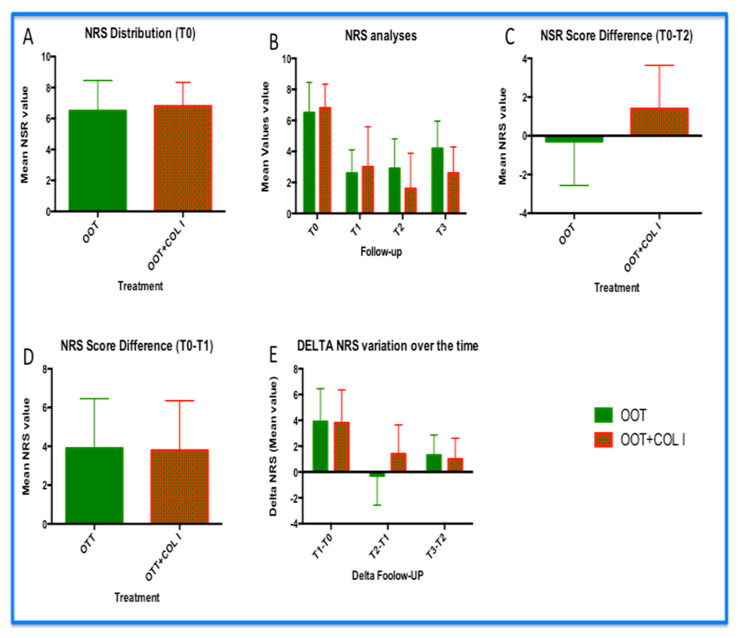
Results of NRS analyses: (**A**) distribution of NRS within the two cohorts of patients, prior to treatments; (**B**) analysis of the mean value of NRS in the follow-up period; (**C**) NSR delta at T1 in both groups; (**D**) change in NRS delta over time; and (**E**) NRS differences between T2 and T1. This graph extrapolation indicates the rate of improvement of the NRS in the two groups.

**Figure 3 diagnostics-14-02411-f003:**
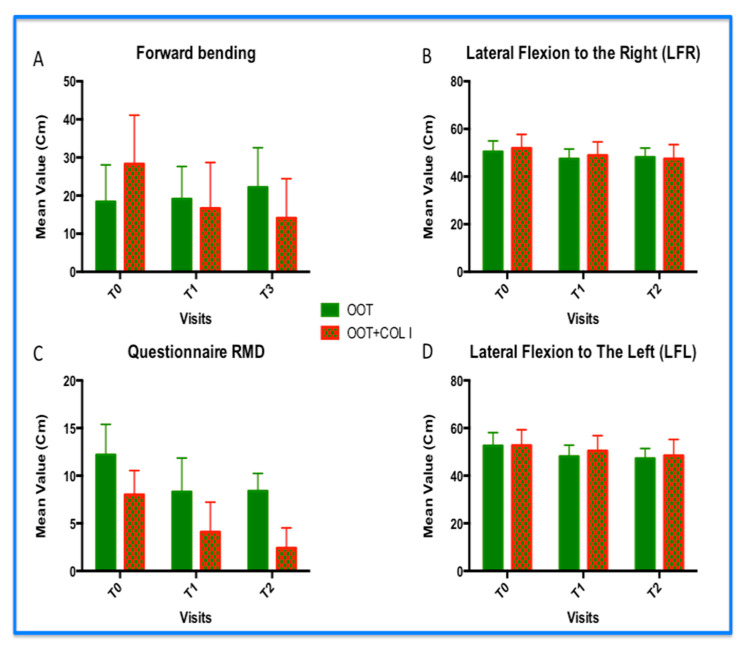
Functional improvement. Results of: (**A**) forward bending; (**B**) lateral flexion to the right (LFR); (**C**) disability questionnaire; and (**D**) lateral flexion to the left (LFR). In graphs (**A**,**C**), the decrease in their average value is statistically significant, where the differences between OOT vs. OOT+COL I are evident (*p* < 0.001). In addition, graph (**A**) shows an anti-parallel direction of the forward bending parameter. Conversely, the values reported in the analysis of the questionnaire revealed the same trend for the treatment with OOT vs. OOT+COL I, but, in the second treatment, the decrease was constant over time compared to that observed in the OOT treatment alone, while, within graphs (**B**,**D**) (lateral motion), the LFR and LFL analyses did not reveal significant differences comparing OOT vs. OOT+COL I treatments. However, the positivity trend was observed in both motions and the result in LFR reached a very close significant value (*p* = 0.0606). In addition, no inferiority level was observed for OOT+COL I compared to OOT treatment alone.

**Figure 4 diagnostics-14-02411-f004:**
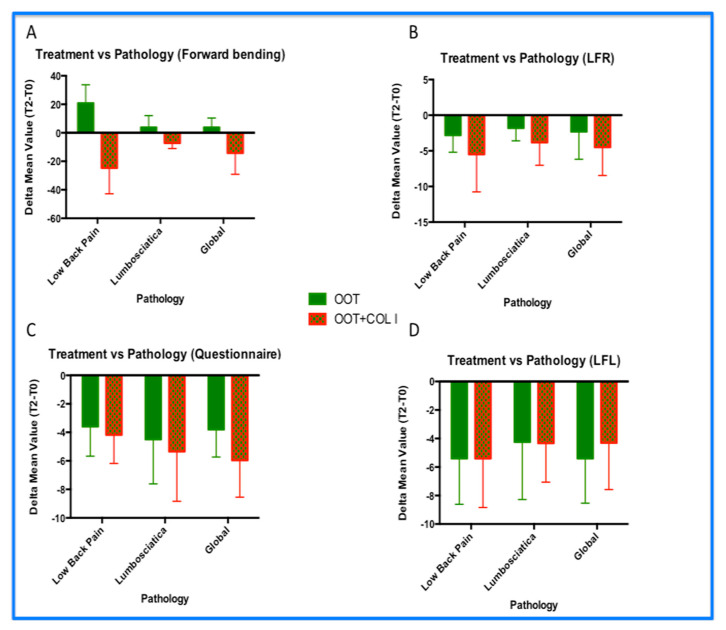
Treatment vs. analysis of pathologies. Results of: (**A**) forward bending; (**B**) lateral flexion to the right (LFR); (**C**) disability questionnaire; and (**D**) lateral flexion to the left (LFL). The results reported the differences between the mean value obtained at the T0 visit and the T2 visit (T2–T0). Negative values represent the positive effects of treatments. In graphs (**A**,**C**), the decrease in their average value is statistically significant (*p* < 0.001), where the differences between OOT vs. OOT+COL I are reversed as shown in Figure 4A. In addition, graph (**A**) shows an anti-parallel parameter of the direction of forward bends. Conversely, the values reported in the questionnaire analyses revealed a progressive positive trend for both treatments; however, in the OOT+COL I treatment, the decrease was significantly higher than that observed in the OOT treatment alone, while, within graphs (**B**,**D**) (lateral motion), the LFR and LFL analyses did not show significant differences comparing OOT vs. OOT+COL I. However, the positivity trend was observed in both movements, and the result in LFR reached a very close significant value (*p* = 0.0606). In addition, no inferiority level was observed for OOT+COL I compared to OOT treatments alone, considering all the parameters analyzed.

**Table 1 diagnostics-14-02411-t001:** Characteristics of treatments. Both treatments were administered over two months.

Group	Treatment	Volume	Number of Treatments
A (10; 5 M and 5 F)	O_2_O_3_	20 ml	8
B (10; 5 M and 5 F)	O_2_O_3_ + MD-LUMBAR	20 mL + 2 mL	8

Note: The number of patients enrolled for each arm is indicated in round brackets.

## Data Availability

All data obtained from this study are available for consultation. The data controller is Liliana Sgarbi.

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
