# Peer review of "Combined Oxygen–Ozone and Porcine Injectable Collagen Therapies Boosting Efficacy in Low Back Pain and Disability"

_diagnostics, 2024, doi:10.3390/diagnostics14212411_

Round 1
Reviewer 1 Report (Previous Reviewer 1)
Comments and Suggestions for Authors
First of all, thanks for the invitation to review this important manuscript.
Of course, the topic of this study is very relevant. Low back pain (LBP) has the highest prevalence globally among musculoskeletal conditions and is the leading cause of disability worldwide. It is the condition where the greatest number of people may benefit from rehabilitation.
The title, in my opinion, must be changed “Oxygen-ozone and porcine collagen injectable therapies improve the quality of life in low back pain patients. Boosting efficacy in pain and disability”. Your study did not examine quality of life. You can remove it and combine the title into one phrase.
In abstract, in lines 21-24: “Results: The analyses revealed significant statistical differences comparing the two arms, where, the OOT COL I treatment demonstrated a booster efficacy in pain, while the questionnaire revealed a reduction in disability. “ Please describe your results in more detail.
In abstract, in lines 24-25: “Therefore, this combination therapy”. Indicate in detail the name of the treatment method.
Add reference to line 35.
In introduction, in lines 31-48: This information can be found in any neurology textbook. Please shorten it.
In introduction: In lines 48-50. “The American Society of Pain and Neuroscience (ASPN) has indicated several procedures and/or treatments including: anti-inflammatory therapy [8], ozone treatment [9–11], minimally invasive procedures [12,13], regenerative drugs [7,14–16] and surgery [17,18]” . This information should be linked to American Society of Pain and Neuroscience (ASPN) references only. If you want to demonstrate the use of multiple treatments that are used to treat LBP, you can start with a new phrase.
Add reference to line 53
In introduction: in lines 55-63: There are 2 references related to the pathogenetic basis of oxygen-ozone infiltration therapy (OIT), dated 2003 and 2013, and one reference without a detailed indication of the source. Recently, many references to this topic have appeared in the literature.
19. Andreula, C.F.; Simonetti, L.; De Santis, F.; Agati, R.; Ricci, R.; Leonardi, M. Minimally Invasive Oxygen-Ozone Therapy for Lumbar Disk Herniation. AJNR Am. J. Neuroradiol. 2003, 24, 996–1000.
20. Rahimi-Movaghar, V.; Eslami, V. The Major Efficient Mechanisms of Ozone Therapy Are Obtained in Intradiscal Procedures. Pain Physician 2012, 15, E1007-1008. 473
21. Rome, Consensus Conference Italian Guidelines and Good Practices in Oxygen-Ozone Therapy 2024.
In introduction: in lines 64-70” Linked references to this important paragraph dated 2005 and 2010. Add more recent references to this paragraph.
Please indicate the approval of the ethical committee and the name of the university to which the ethical committee belongs
Indicate, please, if all procedures were in accordance with the 1984 Declaration of Helsinki and its subsequent amendments, if all patients read the above article in its entirety, including text, figures, and supplementary material, and consented to its publication.
Indicate, please if there was remuneration for participation in this study.
Please indicate whether the participants included patients with rheumatoid diseases, ankylosing spondylitis or gout.
In line 120: What diagnostic criteria were used in the differential diagnosis of LBP and lumbosciatica?
Please indicate which device was used for treatment by OOT?
I think the “Materials and Methods” section will be more readable and understandable if the paragraphs are divided into points.
I think it is incorrect to divide the groups into two subgroups with a minimum sample size of 4 and 5 patients. First, the results may not be significant. Secondly, the sample size you have determined is 18 people, and this should be the minimum number of participants (sample size) in the smallest subgroup.
In lines 242-244: The OOT+COL I treatments increased forward flexion in patients with LBP compared to those with lumbosciatica. Furthemere, OOT+COL I treatment improved forward flexion in both pathologies (p<0.001). Please indicate significant percentage differences in results between groups.
In lines 245 – 247: The OOT+COL I treatments improved better disability condition level in patients with lumbosciatica compared to those with LBP. Again, OOT+COL I treatment improved disability in both pathologies (p<0.01). Please indicate significant percentage differences in results between groups.
In the discussion: lines between 277-310 are repeated in the introduction. Please remove it. Moreover, only the results of your research should be discussed during the discussion.
In lines 366 -368: The self-diffusion capacity of the medical device (MD-LUMBAR) highlights the ability of collagen molecules to enrich the site of interest, where, the EUS-guided injection could be a non-mandatory approach, but remain still recommended. Where can I find this information in results?
In line 369: “maximize the effect of the therapy” Modify please maximize because this effect may not be the maximum that can be achieved with other treatments. What kind of therapy are we talking about here?
In lines 371-372: At the same time, the regenerative properties of porcine collagen type 1 could lead to this treatment being included within the branch of regenerative medicine. Where can I find this information in results?
Paragraph between 373-381 must be removed.
In conclusion: lines 383-384: “Previously, no studies had been conducted regarding the combined use of OOT and porcine collagen type 1 injection” must be removed
In conclusion: lines 384: The combination of these two injective therapies!! indicate which treatment methods are being discussed here in the CONCLUSION!
In conclusion: on line 385 you talk about the safety of the methods used without side effects. However, there is no information about a study unless safety and side effects relate to the methods or results of your study. It would be better if you removed this information or added additional information about safety and adverse effects in the results.
Author Response
|
3. Point-by-point response to Comments and Suggestions for Authors
Of course, the topic of this study is very relevant. Low back pain (LBP) has the highest prevalence globally among musculoskeletal conditions and is the leading cause of disability worldwide. It is the condition where the greatest number of people may benefit from rehabilitation
Thank you for pointing this out. We agree with this comment and we would like to thank the reviewer.
|
|
|
Comment 1: The title, in my opinion, must be changed “Oxygen-ozone and porcine collagen injectable therapies improve the quality of life in low back pain patients. Boosting efficacy in pain and disability”. Your study did not examine quality of life. You can remove it and combine the title into one phrase. |
|
|
Response 1: Thank you for pointing this out. We agree with this comment. Therefore, we have changed the title of manuscript accordingly.
|
|
|
Comment 2: In abstract, in lines 21-24: “Results: The analyses revealed significant statistical differences comparing the two arms, where, the OOT COL I treatment demonstrated a booster efficacy in pain, while the questionnaire revealed a reduction in disability. “ Please describe your results in more detail. |
|
|
|
|
Comments 3: In abstract, in lines 24-25: “Therefore, this combination therapy”. Indicate in detail the name of the treatment method |
|
|
Response 3: Thank you for pointing this out. We agree with this comment. Therefore, we have revised the abstract in order to ameliorate this aspect. Please find our modification in the track changes in red inside the word file.
Comment 4: Add reference to line 35. |
|
|
Response 4: Thank you for pointing this out. We added a reference following your indication.
Comments 5: In introduction, in lines 31-48: This information can be found in any neurology textbook. Please shorten it. |
|
|
Response 5: Thank you for pointing this out. We agree with this comment. Therefore, we have changed the section introduction accordingly. Please find our modification in the track changes in red inside the word file.
Comments 6: In introduction: In lines 48-50. “The American Society of Pain and Neuroscience (ASPN) has indicated several procedures and/or treatments including: anti-inflammatory therapy [8], ozone treatment [9–11], minimally invasive procedures [12,13], regenerative drugs [7,14–16] and surgery [17,18]” . This information should be linked to American Society of Pain and Neuroscience (ASPN) references only. If you want to demonstrate the use of multiple treatments that are used to treat LBP, you can start with a new phrase. |
|
|
Response 6: Agree. We have, accordingly, revised the introduction sections looking for this indication inside the guidelines for LBP management. Please find the track changes in red in the word file.
Comments 7: Add reference to line 53. |
|
|
Response 7: Thank you for pointing this out. We added a reference following your indication.
Comments 8: In introduction: in lines 55-63: There are 2 references related to the pathogenetic basis of oxygen-ozone infiltration therapy (OIT), dated 2003 and 2013, and one reference without a detailed indication of the source. Recently, many references to this topic have appeared in the literature. |
|
|
Response 8: Thank you for pointing this out. We agree with this comment. Therefore, we have changed the references as well you told us above. Please find the track changes in red in the word file.
Comments 9: In introduction: in lines 64-70” Linked references to this important paragraph dated 2005 and 2010. Add more recent references to this paragraph. |
|
|
Response 9: Thank you for pointing this out. We added a reference following your indication.
Comments 10 Please indicate the approval of the ethical committee and the name of the university to which the ethical committee belongs. |
|
|
Response 10: Thank you for pointing this out. Please find the disclosure concerning the ethical committee. This aspect was clarified before with the editorial board of the journal.” Institutional Review Board Statement: For this study, the approval of the ethics committee was not necessary since it is a retrospective analysis on anonymized data. The data in possession for the analyses can be considered anonymized, when it is not possible to trace the identity of the subjects even using subsequent operations at the informatic level. This aspect is regulated by law no. 675/1996 of the Guarantor of Privacy in Italy, in compliance with the use of personal data for scientific purposes.”
Comments 11: Indicate, please, if all procedures were in accordance with the 1984 Declaration of Helsinki and its subsequent amendments, if all patients read the above article in its entirety, including text, figures, and supplementary material, and consented to its publication |
|
|
Response 11: Thank you for pointing this out. Please find the disclosure put in the back materials of present version of manuscript.
Comments 12: Indicate, please if there was remuneration for participation in this study. |
|
|
Response 12: Thank you for pointing this out. There was no remuneration for participation in this study.
Comments 13: Please indicate whether the participants included patients with rheumatoid diseases, ankylosing spondylitis or gout. |
|
|
Response 13: Thank you for this question. No, only data from LBP patients were selected for this analysis.
Comments 14: In line 120: What diagnostic criteria were used in the differential diagnosis of LBP and lumbosciatica? |
|
|
Response 14: Thank you for pointing this out. We extracted this information anonymously. We believe that the doctors who recorded this data used a differential criterion based on pain: whether it was present only in the back, or also present in the branches of the sciatic nerve.
Comments 15: Please indicate which device was used for treatment by OOT? Response 15: Thank you for pointing this out. We described this aspect inside the materials and methods section. Indeed, in order to clarify better this aspect, we prepared also the figure number 1.
|
Comments 16: I think the “Materials and Methods” section will be more readable and understandable if the paragraphs are divided into points.
Response 16: Agree. We have, accordingly, changed the format of materials and methods section. Please find our modification in the track changes in red inside the word file.
Comments 17: I think it is incorrect to divide the groups into two subgroups with a minimum sample size of 4 and 5 patients. First, the results may not be significant. Secondly, the sample size you have determined is 18 people, and this should be the minimum number of participants (sample size) in the smallest subgroup.
Response 17: Thank you for pointing this out. Thank you so much for your point of view. The lumbosciatica subgroups are very small, however, we make an attempt in order to investigate whether the mean big difference observed in our analyses might reflect a significant value in also in a clinical aspect. We added this comment in our manuscript: ”Furthermore, it is It is absolutely necessary to emphasize that the group of patients with lumbosciatica is only 9 in total and although statistical tests revealed statistically significant differences, these results need to be confirmed in a larger cohort of LBP patients with lumbosciatica’.
Comments 18: In lines 242-244: The OOT+COL I treatments increased forward flexion in patients with LBP compared to those with lumbosciatica. Furthemere, OOT+COL I treatment improved forward flexion in both pathologies (p<0.001). Please indicate significant percentage differences in results between groups.
Response 18: Thank you for pointing this out. We reported the percentage of variation in both groups in the main text of manuscript. Please find our modification in the track changes in red inside the word file.
Comments 19: In lines 245 – 247: The OOT+COL I treatments improved better disability condition level in patients with lumbosciatica compared to those with LBP. Again, OOT+COL I treatment improved disability in both pathologies (p<0.01). Please indicate significant percentage differences in results between groups.
Response 19: Thank you for pointing this out. We reported the percentage of variation in both groups in the main text of manuscript. Please find our modification in the track changes in red inside the word file.
Comments 20: In the discussion: lines between 277-310 are repeated in the introduction. Please remove it. Moreover, only the results of your research should be discussed during the discussion.
Response 20: Agree. We have, accordingly, revised the Discussion sections looking for this indication. Please find the track changes in red in the word file.
Comments 21: In lines 366 -368: The self-diffusion capacity of the medical device (MD-LUMBAR) highlights the ability of collagen molecules to enrich the site of interest, where, the EUS-guided injection could be a non-mandatory approach, but remain still recommended. Where can I find this information in results?
Response 21: Thank you for pointing this out. Looking for the literature we found an interesting paper reporting the role of EUS-guided injection (see added reference). The authors demonstrated their best results without US approach in injection treatment of shoulder pathology. Please find inside the text our modifications.
Comments 22: In line 369: “maximize the effect of the therapy” Modify please maximize because this effect may not be the maximum that can be achieved with other treatments. What kind of therapy are we talking about here?
Response 22: Agree. We have, accordingly, revised the Discussion sections looking for this indication. Please find the track changes in red in the word file.
Comments 23: In lines 371-372: At the same time, the regenerative properties of porcine collagen type 1 could lead to this treatment being included within the branch of regenerative medicine. Where can I find this information in results?
Response 23: Agree. We have, accordingly, revised the Discussion sections looking for this indication. We removed this part. Please find the track changes in red in the word file.
Comments 24: Paragraph between 373-381 must be removed.
Response 24; Ok. Thanks for the suggestion.
Comments 25: In conclusion: lines 383-384: “Previously, no studies had been conducted regarding the combined use of OOT and porcine collagen type 1 injection” must be removed
Response 25: Ok. Thanks for the suggestion.
Comments 26: In conclusion: lines 384: The combination of these two injective therapies!! indicate which treatment methods are being discussed here in the CONCLUSION!
Response 26: Thank you for pointing this out. We specified this aspect.
Comments 27: In conclusion: on line 385 you talk about the safety of the methods used without side effects. However, there is no information about a study unless safety and side effects relate to the methods or results of your study. It would be better if you removed this information or added additional information about safety and adverse effects in the results.
Response 27: Thank you for pointing this out. We changed this aspect inside the conclusion section section, following your suggestion.
|
4. Response to Comments on the Quality of English Language |
|
Point 1: I am not qualified to assess the quality of English in this paper. |
|
Response 1: Thank you for pointing this out. We have modified the language of manuscript according the comments for the reviewer number two and the editorial office. |
|
5. Additional clarifications |
|
We haven’t additional clarifications for the editorial office. |

Reviewer 2 Report (Previous Reviewer 2)
Comments and Suggestions for Authors
The authors have made some improvements, but not all of the requested changes have been implemented. Therefore, additional modifications are required before the manuscript can be considered acceptable for publication
1. Please provide a rationale for the statistical analysis, including the use of normality tests (such as the Shapiro-Wilk test, Kolmogorov-Smirnov test, or others) to justify the application of parametric tests, including ANOVA and t-tests. Specifically, clarify how the results of these normality tests (with p-values greater than 0.05) support the assumption of normal distribution, thereby justifying the use of parametric methods.
2. “For this study, the approval of the ethics committee was not necessary since it is a retrospective analysis on anonymized data”
This statement is not entirely accurate. Even retrospective studies using anonymized data may require approval from an ethics committee, particularly to ensure proper data anonymization procedures are followed. However, if the editor accepts the justification provided by the authors, I have no further objections.
3. There is still only 1 reference in: Comment 9: “several studies, in which similar results are reported [1].” Which are the
“several studies”? There is only one reference.
Response 9: Agree. We have, accordingly, modified this part of manuscript. Please find our modification in the track changes in red inside the word file.
4. Same for: Comment 10: “First, different protocols were used in each study, with different concentrations and doses of ozone, routes of application, and methods for evaluating the results [10].” Which are the “each study”? There is only one reference.
Response 10: Thank you for pointing this out. We agree with this comment. We increased the number of references associated with this description. Please find our modification in the track changes in red inside the word file.
5. The authors did not improve the English, and some words were poorly written as requested in:
Comment 14: Additionally, misplaced words like “lumbar bar disc herniation” and
misspelled words such as “collagene,” “assesed,” and “efficay” should be corrected.
Response 14: Thank you for pointing this out. We agree with this comment. This fact is very
important for the meaning of manuscript. We have, accordingly, made the changes and replaced the
wrong words. Please find our modification in the track changes in red inside the word file.
For example: “Nonetheless, the efficay and tolerability of porcine 94 collagene type I was assesed previously [31,32].”
Comments on the Quality of English LanguageThe authors did not improve the English, and some words were poorly written as requested in:
Comment 14: Additionally, misplaced words like “lumbar bar disc herniation” and
misspelled words such as “collagene,” “assesed,” and “efficay” should be corrected.
Response 14: Thank you for pointing this out. We agree with this comment. This fact is very
important for the meaning of manuscript. We have, accordingly, made the changes and replaced the
wrong words. Please find our modification in the track changes in red inside the word file.
For example: “Nonetheless, the efficay and tolerability of porcine 94 collagene type I was assesed previously [31,32].”
Author Response
|
3. Point-by-point response to Comments and Suggestions for Authors
The authors have made some improvements, but not all of the requested changes have been implemented. Therefore, additional modifications are required before the manuscript can be considered acceptable for publication. Thank you for pointing these aspects out.
|
|
|
Comment 1: Please provide a rationale for the statistical analysis, including the use of normality tests (such as the Shapiro-Wilk test, Kolmogorov-Smirnov test, or others) to justify the application of parametric tests, including ANOVA and t-tests. Specifically, clarify how the results of these normality tests (with p-values greater than 0.05) support the assumption of normal distribution, thereby justifying the use of parametric methods |
|
|
Response 1: Thank you for pointing these aspects out. We agree with this comment and we performed the Shapiro-Wilk test in order to understand whether we have a normal distribution of data. Here, we are reporting the result concerning the both W and p values for the OOT group (W=0.9066 and p=0.2646) and OOT+COL I group (W=0.9175 and p=0.3476, respectively. We completed this part in materials and methods section.
|
|
|
Comment 2: “For this study, the approval of the ethics committee was not necessary since it is a retrospective analysis on anonymized data” .This statement is not entirely accurate. Even retrospective studies using anonymized data may require approval from an ethics committee, particularly to ensure proper data anonymization procedures are followed. However, if the editor accepts the justification provided by the authors, I have no further objections. |
|
|
|
|
Comment 3: P There is still only 1 reference in: Comment 9: “several studies, in which similar results are reported [1].” Which are the“several studies”? There is only one reference. Response 9: Agree. We have, accordingly, modified this part of manuscript. Please find our modification in the track changes in red inside the word file. |
|
|
Response 3: Thank you for pointing this out. We agree with this comment. Therefore, we have removed this part according the suggestion of reviewer number 1. Please find our modification in the track changes in red inside the word file.
|
|
|
Comment 4: Same for: Comment 10: “First, different protocols were used in each study, with different concentrations and doses of ozone, routes of application, and methods for evaluating the results [10].” Which are the “each study”? There is only one reference. Response 10: Thank you for pointing this out. We agree with this comment. We increased the number of references associated with this description. Please find our modification in the track changes in red inside the word file. |
|
|
Response 4: Thank you for pointing this out. We agree with this comment. . Therefore, we havemodified this part according also the suggestion of reviewer number 1. Please find our modification in the track changes in red inside the word file.
|
|
|
Comment 5: The authors did not improve the English, and some words were poorly written as requested in: Comment 14: Additionally, misplaced words like “lumbar bar disc herniation” and misspelled words such as “collagene,” “assesed,” and “efficay” should be corrected. Response 14: Thank you for pointing this out. We agree with this comment. This fact is very important for the meaning of manuscript. We have, accordingly, made the changes and replaced the wrong words. Please find our modification in the track changes in red inside the word file. For example: “Nonetheless, the efficay and tolerability of porcine 94 collagene type I was assesed previously [31,32].” Response 5: Thank you for pointing this out. In previous revision we received the following indication: ”Moderate editing of English language required”. We increased again the revision of manuscript in order to avoid typing errors. Please find our modification in the track changes in red inside the word file.
|
|
|
4. Additional clarifications |
|
|
We haven’t additional clarifications for the editorial office. |

Round 2
Reviewer 1 Report (Previous Reviewer 1)
Comments and Suggestions for Authors
As I noted earlier, the topic is very relevant and it is necessary to continue studying this topic in this direction.
In general, the authors have done an excellent work in revising this manuscript based on the comments from the reviewers. Most of my earlier concerns have been addressed by the authors in this revised version. The manuscript is now situated in context, with appropriate referencing, and the novelty is clearer. The introduction, materials and methods, results, discussion and conclusion are significantly better makes it a more complete paper, suitable for publication in this high rating journal.
Reviewer 2 Report (Previous Reviewer 2)
Comments and Suggestions for Authors
The authors have addressed most of the requested revisions. However, there is a lack of attention to detail in writing a scientific manuscript, particularly with correct punctuation and other stylistic elements. Additionally, there are some grammatical errors, such as the use of "collagene" instead of the correct English term "collagen." While "collagene" is appropriate in French or Italian, in English, "collagen" should be used. Despite these issues, I have no further suggestions.
Comments on the Quality of English LanguageWhile "collagene" is appropriate in French or Italian, in English, "collagen" should be used.
This manuscript is a resubmission of an earlier submission. The following is a list of the peer review reports and author responses from that submission.
Round 1
Reviewer 1 Report
Comments and Suggestions for Authors
This topic is widespread due to the growing incidence of LBP in all age groups.
The title should not contain an abbreviation. Moreover, the title should be specific and understandable. What kind of efficiency are we talking about here? analgesic? recovery? regenerative?
In introduction, too much attention is paid to the anatomy and physiology of connective tissue and its role in the formation of pain.
10 patients in each group is too small a sample size. I think the small sample size may affect the significance of your results. However, indicate the method you used to determine the sample size.
Indicate the number of patients evaluated for eligibility, after randomization and the number of excluded patients and the reason for their exclusion.
In the inclusion criteria, indicate the maximum size of the disc herniation; were there any indications for surgical intervention?
in inclusion criteria specify the level of pain in studied groups.
There is no description of assessment methods in materials and methods.
There is no description for selecting injection points. Indicate how did you select these points?
3.3. Pathology of machining. This section is written in incomprehensible language. What did the authors mean by pathologies and treatments?
I don't understand why the authors only included patients with herniated discs. What did they want to prove with this? However, MRI after treatment was not performed to compare size dynamics after treatment. I hope this will be discussed in detail in the discussion.
In the discussion it is necessary to explain the secret of the effectiveness of this method.
The conclusion should be short and clear. written in a few sentences and ending with a recommendation for use and requirements for continued research in this direction.
Add reference in line 71 after “system”
Add reference in line73 after “body”
Add reference in line 75 after “system”
Add reference in line 81 after “system”
Add reference in line 83 after “tissue”
Add reference in line 85 after “contraction”
Add reference in line 87 after “tension”
Add reference in line 93 after “ECM”
Add reference in line 97 after “phenomena”
Add reference in line102 after “charge”
Add reference in line 109 after “subject”
Add reference in line 110 after “fibrosis”
Add reference in line 118 after “it self”
Add reference in line 120 after “phenomena”
Add reference in line 122 after “fibers”
Add reference in line 110 after “fibrosis”
Add reference in line 118 after “it self”
What is the purpose of demonstrating the herniated disc in Figure 1? 1. Please cite the source and provide a detailed description of the MRI.
Tables and figures are not designed according to the rules of the journal
Author Response
For research article
|
Response to Reviewer 1 Comments
|
|||
|
1. Summary |
|
|
|
|
Thank you very much for taking the time to review this manuscript. Please find the detailed responses below and the corresponding revisions/corrections highlighted/in track changes in the re-submitted files.
|
|||
|
2. Questions for General Evaluation |
Reviewer’s Evaluation |
Response and Revisions |
|
|
Does the introduction provide sufficient background and include all relevant references? |
Must be improved |
Please find our replies inside this document, below. |
|
|
Are all the cited references relevant to the research? |
Must be improved |
|
|
|
Is the research design appropriate? |
Must be improved |
|
|
|
Are the methods adequately described? |
Must be improved |
|
|
|
Are the results clearly presented? |
Must be improved |
|
|
|
Are the conclusions supported by the results?
|
Must be improved |
|
|
|
3. Point-by-point response to Comments and Suggestions for Authors |
|||
|
Comment 1: This topic is widespread due to the growing incidence of LBP in all age groups. |
|||
|
Response 1: Thank you for pointing this out. We agree with this comment. Therefore, we have mentioned this fact at the beginning of the abstract and inside the introduction section.
|
|||
|
Comment 2: The title should not contain an abbreviation. Moreover, the title should be specific and understandable. What kind of efficiency are we talking about here? analgesic? recovery? regenerative? |
|||
|
|||
|
Comments 3: In introduction, too much attention is paid to the anatomy and physiology of connective tissue and its role in the formation of pain groups. |
|||
|
Response 3: Thank you for pointing this out. We agree with this comment. Therefore, we have revised the chapter of introduction in order to ameliorate this aspect, reducing the part of anatomy and physiology. Please find our modification in the track changes in red inside the word file
Comment 4: 10 patients in each group is too small a sample size. I think the small sample size may affect the significance of your results. However, indicate the method you used to determine the sample size. |
|||
|
Response 4: Thank you for pointing this out. We would like to reply this comment. We evaluate the sample dimension by the appropriate formula. We set the power of the study at 80% and we have been considered the percent of the Adverse event equal to 15% (considering both injection). Indeed, the result of formula (N=Log(1-85%)/Log(1-10%), is N=18. For these reasons, we expected that the number of 10 patients per arm (sum 20) was sufficient. Therefore, we have mentioned this fact at the materials and methods section.
Comments 5: Indicate the number of patients evaluated for eligibility, after randomization and the number of excluded patients and the reason for their exclusion. |
|||
|
Response 5: Thank you for pointing this out. We agree with this comment. Therefore, we have changed the section of material and methods, indicating the total number of patients evaluated for elegibility and the exclusion criteria. Considering both the sample power and inclusion criteria, a total of 20 patients were included in the study.
Comments 6: In the inclusion criteria, indicate the maximum size of the disc herniation; were there any indications for surgical intervention? |
|||
|
Response 6: Agree. We have, accordingly, revised the materials and method sections looking for this indication inside the guidelines for LBP management. Please find the track changes in red in the word file.
Comments 7: In inclusion criteria specify the level of pain in studied groups. |
|||
|
Response 7: Thank you for pointing this out. We agree with this comment. Therefore, we have insert this aspect in the material and methods section. Please find the track changes in red in the word file.
Comments 8: There is no description of assessment methods in materials and methods. |
|||
|
Response 8: Thank you for pointing this out. We agree with this comment. Therefore, we have changed the materials and methods section following our comments 6,7 and 8. Please find the track changes in red in the word file.
Comments 9: There is no description for selecting injection points. Indicate how did you select these points? |
|||
|
Response 1: Thank you for pointing this out. We agree with this comment. Therefore, we have mentioned this fact at the materials and methods section. Nevertheless, we insert the MRI image inside the Figure 1 (ex Figure 2) in order simplify the injection point explanation. Please find the track changes in red in the word file.
Comments 10: 3.3. Pathology of machining. This section is written in incomprehensible language. What did the authors mean by pathologies and treatments? |
|||
|
Response 10: Thank you for pointing this out. We agree with this comment. Therefore, we have changed this section in order to clarify for the readers. Several sentences were changed. Please find the track changes in red in the word file.
Comments 11: I don't understand why the authors only included patients with herniated discs. What did they want to prove with this? However, MRI after treatment was not performed to compare size dynamics after treatment. I hope this will be discussed in detail in the discussion. |
|||
|
Response 11: Thank you for pointing this out. The use of ozone therapy makes sense only in this category of patients. The demonstration of this condition has been documented through MRI images. If after the treatment no painful conditions are highlighted for the patient, there are no indications to perform MRI investigations according to the indications of the Italian national health system. Please find the track changes in red inside the discussion section of manuscript (word file).
Comments 12: In the discussion it is necessary to explain the secret of the effectiveness of this method. |
|||
|
Response 12: We agree with this comment. Therefore, we have change the discussion reporting our personal consideration about this fact. Please find the track changes in red inside the discussion section of manuscript (word file).
Comments 13: The conclusion should be short and clear. written in a few sentences and ending with a recommendation for use and requirements for continued research in this direction. |
|||
|
Response 13: Thank you for pointing this out. We agree with this comment. Therefore, we have completely changed the conclusion according your suggestions. Please find the track changes in red in the word file.
Comments 14: Add reference in line 71 after “system”; Add reference in line73 after “body”; Add reference in line 75 after “system”; Add reference in line 81 after “system”; Add reference in line 83 after “tissue”; Add reference in line 85 after “contraction”; Add reference in line 87 after “tension”; Add reference in line 93 after “ECM”; Add reference in line 97 after “phenomena”; Add reference in line102 after “charge”; Add reference in line 109 after “subject”; Add reference in line 110 after “fibrosis”; Add reference in line 118 after “it self”; Add reference in line 120 after “phenomena”; Add reference in line 122 after “fibers”. |
|||
|
Response 14: Agree. We have, accordingly, changed completely the list of references based on the introduction modification as well as we did, following the indications of comment three. Please find our modification in the track changes in red inside the word file.
Comments 15: What is the purpose of demonstrating the herniated disc in Figure 1? 1. Please cite the source and provide a detailed description of the MRI. Response 15: Thank you for pointing this out. We agree with this comment. Therefore, we have put the figure 1 inside the figure 2, in order to use it to demonstrate the presence of interested pathology. The MRI figure was acquired at T0 visit from patient 5B. This is an original image of MRI. Therefore, we put this modification inside the manuscript. Please find our modification in the track changes in red inside the word file.
|
|||
Comments 16: Tables and figures are not designed according to the rules of the journal.
Response 16: Agree. We have, accordingly, changed the format of figure. We produced our images in TIFF format in high resolution.
|
4. Response to Comments on the Quality of English Language |
|
Point 1: I am not qualified to assess the quality of English in this paper. |
|
Response 1: Thank you for pointing this out. We have modified the language of manuscript according the comments for the reviewer number two and the editorial office. |
|
5. Additional clarifications |
|
We haven’t additional clarifications for the editorial office. |

Reviewer 2 Report
Comments and Suggestions for Authors
Dear Editor and Authors,
The present manuscript addresses a relevant clinical issue by exploring a novel combination therapy for LBP, which is highly prevalent and costly to manage. However, it lacks scientific rigor in writing, and the MDPI template was not followed, as evidenced by the use of different colors and font sizes. There are several mistakes in the text, such as in the example “lumbar bar disc herniation.” The study presents an interesting and potentially valuable combination therapy for LBP, but the small sample size and other methodological limitations necessitate further research before this treatment approach can be recommended in clinical practice. The authors should consider changing the title to reflect a proof-of-concept study.
The paper does not mention whether ethical approval was obtained, nor does it discuss patient consent in detail, which is a critical aspect of clinical research. The abstract is too long and resembles a manuscript introduction. Therefore, it must be rewritten using approximately 200 words, following the abstract structure: background (1 or 2 statements), objectives, methods, results, and conclusions. Remove abbreviations from the keywords.
Avoid being personal in the abstract, such as “We know that the main functional...”; instead, use more scientific language. In the introduction, again, avoid personal language like “We know that the collagen composition of the ECM is not fixed but that it depends on the stress to which it is subjected.” Instead, write: “It has been demonstrated that the collagen composition of the ECM is not stable...”.
Please clearly state the objectives of the study using scientific language.
Justify how the sample size was calculated and discuss the power of the study.
Clearly describe how patients were recruited, how long the recruitment took, how many dropouts occurred, and the reasons for these dropouts.
Please add a legend with the acronyms used in the figures and tables. Improve the resolution of the images; for instance, Figure 3 is distorted and needs to be enhanced.
The discussion section is brief and lacks depth. Furthermore, there are only a few references. To strengthen this section, consider comparing the proposed approach with other approaches for a broader landscape of LBP management, including up-to-date published manuscripts in the area.
“The literature emphasizes the efficacy and safety of ozone therapy for low back pain (LBP) due to lumbar bar disc herniation.” Which literature? There are no references provided here.
“Most articles have shown that OOT treatment improved outcomes in both pain and functional status with treatment, including ozone vs. ozone-free group (controls) [11].” Which ones are the “Most articles”? There is only one reference.
“several studies, in which similar results are reported [1].” Which are the “several studies”? There is only one reference.
“First, different protocols were used in each study, with different concentrations and doses of ozone, routes of application, and methods for evaluating the results [10].” Which are the “each study”? There is only one reference.
Add a paragraph reporting all the limitations of the study at the end of the discussion section. For example: The study's small sample size (20 patients) limits the generalizability of the findings…. While the study includes a six-month follow-up, longer-term effects beyond this period are not explored, which is critical for understanding the durability of the treatment effects….
The study does not mention whether blinding was implemented, which could introduce bias in the outcome assessments. Potential confounding factors, such as differences in patients' baseline characteristics or concurrent therapies, could also influence the results.
The conclusions are too long, and part of them should be moved to the discussion section.
Comments on the Quality of English LanguageEnglish must be reviewed by a native speaker. There are several nonsense sentences such as:
“A total of 20 patients (10 males and 10 females) were closed to receive treatments.”
“In which, the dual choice (OOT+COL I), demonstrated a booster effect in both short-term and long-term NSR (p<0.0001).”
Additionally, misplaced words like “lumbar bar disc herniation” and misspelled words such as “collagene,” “assesed,” and “efficay” should be corrected.
Author Response
For research article
|
Response to Reviewer 2 Comments
|
|||
|
1. Summary |
|
|
|
|
Thank you very much for taking the time to review this manuscript. Please find the detailed responses below and the corresponding revisions/corrections highlighted/in track changes in the re-submitted files.
|
|||
|
2. Questions for General Evaluation |
Reviewer’s Evaluation |
Response and Revisions |
|
|
Does the introduction provide sufficient background and include all relevant references? |
Must be improved |
Please find our replies inside this document, below. |
|
|
Are all the cited references relevant to the research? |
Must be improved |
|
|
|
Is the research design appropriate? |
Must be improved |
|
|
|
Are the methods adequately described? |
Must be improved |
|
|
|
Are the results clearly presented? |
Must be improved |
|
|
|
Are the conclusions supported by the results?
|
Must be improved |
|
|
|
3. Point-by-point response to Comments and Suggestions for Authors |
|||
|
Comment 1: The present manuscript addresses a relevant clinical issue by exploring a novel combination therapy for LBP, which is highly prevalent and costly to manage. However, it lacks scientific rigor in writing, and the MDPI template was not followed, as evidenced by the use of different colors and font sizes. There are several mistakes in the text, such as in the example “lumbar bar disc herniation.” The study presents an interesting and potentially valuable combination therapy for LBP, but the small sample size and other methodological limitations necessitate further research before this treatment approach can be recommended in clinical practice. The authors should consider changing the title to reflect a proof-of-concept study. |
|||
|
Response 1: Thank you for pointing these aspects out. We agree with this comment, specially concerning the language and the template used. We downloaded the template again from MDPI journal and we paste the font and the size in our revised manuscript. Thank you so much to point out the sample size in our study. Furthermore, we have evaluate the sample size by the power study formula end this result is very close our sampling selection. We described the power of study in materials and methods section. Please find our modification in the track changes in red inside the word file.
|
|||
|
Comment 2: The paper does not mention whether ethical approval was obtained, nor does it discuss patient consent in detail, which is a critical aspect of clinical research. The abstract is too long and resembles a manuscript introduction. Therefore, it must be rewritten using approximately 200 words, following the abstract structure: background (1 or 2 statements), objectives, methods, results, and conclusions. Remove abbreviations from the keywords. Avoid being personal in the abstract, such as “We know that the main functional...”; instead, use more scientific language. In the introduction, again, avoid personal language like “We know that the collagen composition of the ECM is not fixed but that it depends on the stress to which it is subjected.” Instead, write: “It has been demonstrated that the collagen composition of the ECM is not stable...”. |
|||
|
|||
|
Comment 3: Please clearly state the objectives of the study using scientific language. |
|||
|
Response 3: Thank you for pointing this out. We agree with this comment. Therefore, we have mentioned this fact in the abstract and at the end of introduction section. Please find our modification in the track changes in red inside the word file.
|
|||
|
Comment 4: Justify how the sample size was calculated and discuss the power of the study. Clearly describe how patients were recruited, how long the recruitment took, how many dropouts occurred, and the reasons for these dropouts. |
|||
|
Response 4: Thank you for pointing this out. We agree with this comment. Therefore, we have modified all these aspects in the materials and methods section. Please find our modification in the track changes in red inside the word file.
|
|||
|
Comment 5: Please add a legend with the acronyms used in the figures and tables. Improve the resolution of the images; for instance, Figure 3 is distorted and needs to be enhanced. |
|||
|
|||
|
|||
|
4. Response to Comments on the Quality of English Language |
|||
|
Point 1: Moderate editing of English language required. |
|||
|
Response 1: Thank you for pointing this out. We have modified the language of manuscript according the comments for the reviewer following the indication provided by the comments 15 and 16. |
|||
|
5. Additional clarifications |
|||
|
We haven’t additional clarifications for the editorial office. |
|||
